# The Influence of Multiple Types of Flexible Resources on the Flexibility of Power System in Northwest China

Jun Dong, Zhenjie Chen *[image ID] and Xihao Dou

Department of Economic Management, North China Electric Power University, Beijing 102206, China
* Correspondence: 120202206267@ncepu.edu.cn

**Abstract:** The development of renewable energy is of great significance to relieve the pressure on the energy supply and promote the low-carbon operation of the power system. However, the volatility of renewable energy, especially wind and solar energy, has a great impact on the safe and reliable operation of the power system. If we want to introduce renewable energy and ensure the safe and reliable operation of the power system, it is necessary for the power system to provide enough flexibility. Northwest China is rich in natural resources and an important area of power supply in China, which also faces the problem of insufficient flexibility. Therefore, based on the power system development and natural conditions in northwest China, this paper studies the key factors affecting the flexibility of the power system when renewable energy accounts for a large proportion, and proposes measures to improve the flexibility of the power system by using the power system optimization tool Flextool developed by IRENA.

**Keywords:** flexible resources; power system flexibility; power system optimization; flextool

## 1. Introduction

### 1.1. Background

With the massive exploitation and use of non-renewable energy, many countries around the world are facing the dilemma of an energy crisis [1]. At the same time, with the increase in greenhouse gas emissions and the worsening of global environmental problems, as well as the commitments made by countries to carbon peaks and carbon neutrality, renewable energy power generation has gradually attracted wide attention from various countries. Among them, renewable energy represented by solar energy and wind energy develops rapidly. The development and utilization of renewable energy is also an important way for the power sector to achieve low-carbon and clean development, which is conducive to the sustainable development of production and living activities [2]. However, renewable energy, such as wind and solar energy, is greatly affected by the environment, and the uncertainty and intermittence of its power generation have brought huge challenges to the power system's safe operation.

In order to accommodate more renewable energy and at the same time guarantee the security of the power system, each department takes some measures to improve the regulating ability of the power system, such as thermal power unit transformation, increasing the energy storage, and increasing the channel capacity between nodes, but power systems in different regions have different characteristics, as the key region of China's energy output, it is of great significance to study the flexible transformation measures of the power system in northwest China to maintain the stability of its power system.

### 1.2. Literature Review

China's power sector is one of the world's largest sources of carbon emissions, and the country has pledged to peak its carbon emissions by 2030 and become carbon neutral by 2060. In recent years, with the rapid growth of energy demand, renewable energy

represented by solar energy and wind energy is also developing rapidly. The development and utilization of renewable energy provide an important way for the power sector to realize low-carbon and clean, and also help alleviate the energy crisis. By 2021, China's installed capacity for renewable energy exceeded 1 billion kW, and the installed capacity of wind power and photovoltaic (PV) power both exceeded 300 million kW. China has also set a target of installing 1200 GW of solar and wind power by 2030 [3].

The development of renewable energy can effectively alleviate the energy crisis and promote the low-carbon process of the power system, but also have an impact on the safe and stable operation of the power system [4]. Due to the access to renewable energy, the energy structure of the power system is changed, and various departments of the power system usually adopt a variety of flexible resources and joint regulation measures, such as improving the rapid climbing start-stop capacity of thermal power units, energy storage, and multi-network interconnection, to cope with the fluctuations of renewable energy [5]. More detailed on the energy supply side of the power system, the research of international scholars is more focused on improving the performance of thermal power units, rationally using virtual synchronous machines and other ways to strengthen the flexibility of the power system, including Lu Q. et al. [6] taking 300 MW and 200 MW cogeneration unit as an example, by analyzing that the hot water storage device can increase the peak-load capacity of the unit from 16% to 37% and from 13% to 27%, respectively, the possibility of improving the flexibility of the power system by improving the capacity of peak regulation of the unit is verified. Jaber A. et al. [7] used the swing equation of synchronous motor to express the virtual inertia characteristics in their research on VSG design and could control the swing equation parameters of VSG in real time to enhance the fast response of the virtual machine tracking steady frequency. This study improves the ability of the power system to resist disturbance. On the grid side of the power system, the measures to improve the flexibility of the power system are mainly to improve the transmission capacity of the power network, including the adoption of Uhv AC-DC(Ultra-high voltage Alternating Current/Direct Current) technology and flexible HVDC (High Voltage Direct Current) technology, such as Zhou J. et al. [8] made a comparative study of UHV AC and DC technologies and proposed the applicability of each technology, which is conducive to the construction planning of improving the flexibility of power grid in various regions. On the load side, the way to improve the flexibility of the power system is to increase the flexibility load, including the application of virtual power plants, multi-energy complementarity, and "vehicle network coordination" of EVs (electric vehicles), etc. In this regard, Li J. et al. [9] study the market mechanism of virtual power plants participating in peak and frequency regulation and sum up the experience of various countries. Xu H. et al. [10] put forward a comprehensive electricity/heat demand response mechanism based on multi-energy complementing, which can tap the response potential of users and achieve a win-win situation for both the power grid and users. Sun J. et al. [11] study the influence of EV charging and discharging behavior on power grid load variation rules under natural charging and orderly charging strategies, and proposed a demand-side management strategy for EVs, which can be better used for peak cutting and valley filling. In terms of energy storage, an energy storage device converts electrical energy into mechanical energy, chemical energy, and other forms to achieve energy storage as an important way to improve the power system's flexibility. Energy storage can be placed in each link of electricity production and transmission, such as power supply, grid and load, it can realize the imbalance regulation of electricity on multiple time scales, and undertake many auxiliary tasks such as peak regulation, frequency modulation, consumption of renewable energy and seasonal electricity balance. Yuan B. et al. [12] propose the application scenarios of energy storage in power supply, power grid, and user side, showing the role of energy storage in improving the flexibility of the power system. Sperstad I.B. et al. [13] account for uncertainties due to distributed wind and solar photovoltaic power generation beyond the planning horizon in an AC (alternating current) MPOPF (multi-period optimal power flow) model for distribution systems with energy storage systems and prove the effectiveness of the strategy. Hosseini S.M. et al. [14]

develop a robust optimization framework for the day-ahead energy scheduling of a grid-connected residential user and confirm that scheduling the RES (renewable energy source) and ESS (energy storage system), power grids are becoming more secure and efficient in the electricity market. Moreover, Microgrids are often used to facilitate the use of distributed energy, and then improve the capacity of the power system to receive renewable energy. Raffaele C. et al. [15] built a multi-carrier microgrid with an energy storage system, adopted an innovative RMPC algorithm and proposed a modeling framework for a microgrid energy management system, including heat and electricity, to solve the interference of uncertainty in the system model. Li Q. et al. [16] put forward a kind of electric–hydrogen hybrid energy storage based on DP-MPC micro grid real-time energy management method, this method can allocate batteries, fuel cells, cell energy, and the network, the output of distributed power supply can be maximally realized while ensuring the power balance and cost optimization of the system.

It can be seen that there are many ways to improve the flexibility of the power system from the power source, power grid, load, and storage. Therefore, many scholars have studied the effects of various ways to improve the flexibility of the power system. Generally speaking, it includes the following two types. The first one is to construct the power system flexibility index to evaluate the operating characteristics of the system. Such as "IRRE" (insufficient ramping resource expectation) and "TUSFI-TEUSFI" (Technical Uncertainty Scenarios Flexibility Index-Technical Economic Uncertainty Scenarios Flexibility Index) proposed by Xiao D. et al. [17] respectively represent the probability expectation that the power system cannot cope with the load change of the grid and the flexibility change of the system caused by the power flow capacity change. Also, the flexibility margin distribution and the probability of insufficient flexibility proposed by Lu Z. et al. [18] establish the quantitative relationship between flexibility, the level of renewable energy consumption, and the risk of load loss. The second is dynamic simulation evaluation, that is, several scenarios are set to optimize scheduling and production simulation, and the flexibility of the system is often judged according to whether there is a load loss or a renewable energy power limit. Tang X. et al. [19] establish a multi-time scale optimization scheduling strategy considering multi-energy flexibility, it can effectively improve the flexibility of the system without significantly reducing the economic efficiency. Hussam N. et al. [20] establish the system flexibility demand and supply model under the time scale of 1 h, and analyzed the system flexibility supply and demand balance in the short-term operational planning.

However, most of the current research on the flexibility of power system focuses on the micro level, that is, studying a specific measure to improve the flexibility of the power system or optimizing the operation of the constructed micro power system. Few of them can expand the research scope to the regional level and carry out the simulation of the real power system. In this paper, the Flextool model is used to simulate the system flexibility changes when the power system in Northwest China changes the status of different flexible resources through hourly system scheduling and operation simulation at the regional level.

The main contributions of this paper include the followings:

1. Simulate the structural characteristics of the power system in a region in northwest China, divide different node setting power transmission channels, and set thermal power units with different technical characteristics, so as to understand the changes brought by the flexible transformation of thermal power units.
2. Simulate the wind power and solar power output scenario, increase the uncertainty of the power system operation, and change the demand of the power system for flexible resources.
3. Set up different power system operation scenarios, compare the benefits brought by different types of power system flexible transformation measures, and analyze more pertinency which way is more suitable to increase system flexible resources in northwest China.

4. This paper takes a certain region in northwest China as a representative, which can provide a reference for the transformation of power systems in the same type of region.

## 2. Strategy and Modeling

Studies evaluating the impact of renewable energy penetration on the power system have used a variety of optimization tools, including production cost models, capacity expansion models, or a combination of these models. The capacity expansion model combines the fixed and variable costs of existing and planned generation, storage, demand-side resources, and transmission infrastructure to select the optimal portfolio of assets to meet electricity demand for years to come. The production cost model only simulates the variable cost of a given generation mix and transmission capacity to meet the minimum cost of electricity demand. Typically, capacity expansion models optimize systems for many years, have low temporal resolution, and describe power systems in less detail. In contrast, production cost models have higher time resolution (minutes to hours) and more detailed descriptions of power systems, but typically only simulate one year of system operation. The tool adopted in this paper is the production cost model.

In this paper, Flextool is used as the research tool, which is the flexibility evaluation software developed by IRENA (International Renewable Energy Agency, https://www.irena.org/, accessed on 1 August 2022), the version used in this paper is version 2.0 (April 2020). It can be used to study system capacity expansion and scheduling simultaneously. Flextool is designed to analyze not only traditional concepts of flexibility (such as cogeneration and hydropower units with high flexibility), but also innovative flexibility technologies such as flexible resources requirements, energy storage, and flexible resources coupling. On the one hand, Flextool analyzes system operations; on the other hand, it optimizes the generation portfolio to minimize costs, and provides a solution for coordinated operation of grid, storage, and demand-side flexible resources. Flextool is data-driven software that feeds demand, generation mix, hydrological data, renewable energy time series, grid information, fuel costs, and more. This paper mainly studies the flexibility of the power system in a certain region in northwest China, the proportion of hydroelectric power generation in this region is very small, and for the convenience of research, this paper only studies the power system including thermal power, wind power and photovoltaic, and does not consider the coupling with other energy networks. Flextool has strong applicability to it. Based on the development status, renewable energy output, and energy structure parameters in this region, this paper simulated the model with the operation data, evaluated the flexibility indexes under different simulation strategies, and analyzed the flexibility changes brought by different strategies. The main input variables used in this article can be summarized as Table 1.

**Table 1.** The description of the main input variables.

| Variable Categories | Description | Relevant Input Variable | Description |
|---|---|---|---|
| Grid | One network for one product, however, only the electric grid is studied in this paper. | Channel | Interconnection status between nodes |
| | | Channel capacity | The capacity of each channel |
| | | Loss | The line loss rate of each power transmission channel |
| Node | The grid is divided into three nodes | Coal-fired unit | Each node has different types of power supply. In this paper, these energy sources only address electrical devices |
| | | Wind power | |
| | | Photovoltaic | |
| | | storage | |

**Table 1.** *Cont.*

| Variable Categories | Description | Relevant Input Variable | Description |
|---|---|---|---|
| Unit type | Different types of power supplies have different performance parameters | Number | The number of the unit in this power system |
| | | Rate of climb | The rate at which the output of a unit increases or decreases per unit time |
| | | Minimum technical output | The minimum load rate of different types of units that can operate safely and stably |
| | | Efficiency | The work efficiency of unit |
| | | Start-stop cost | The cost caused by unit startup and shutdown |
| | | kWh Cost | The cost required by a unit to generate 1 kWh of electricity |
| Timestep | Timesteps are an ordered series of timesteps | Load | Load energy consumption |
| | | Wind power | Output status of wind power |
| | | PV | Output status of solar power |

The flexibility evaluation indicators used in this paper mainly include the following three:

(A). Loss of Load

Loss of load occurs when the supply cannot match the demand and energy must go unserved. Flextool shows the maximum amount of loss of load (MW) given in a single period and the total loss of load (%) of demand.

(B). Curtailment

Curtailment occurs when VRE (variable renewable energy) output has to be reduced because of inflexibility or because VRE generation exceeds the load demand. Flextool measures maximum power curtailed in MW and total power curtailment (%) in the given period.

(C). Reserve Inadequacy

Reserve inadequacy occurs when the reserve requirement in the node cannot be met. Flextool provides the reserve inadequacy as maximum MW for the given period.

Based on those results, on the one hand, can horizontal contrast the difference between different flexibility improvement measures, on the other hand, the influence degree of different strategies on system flexibility improvement can be analyzed longitudinally, it has high operability. However, it is worth mentioning that because this tool uses a linear programming method, it cannot establish a binary decision model and carry out the ultra-short-term simulation.

## 3. Benchmark Power System Model Construction

### 3.1. Power System Base Scenario Model Construction

In a province in northwest China, by the end of 2020, the total installed capacity of power grid integrated adjustment was 55,869.913 MW, including 30 thermal power plants and 66 units with a capacity of 29,710.4 MW, accounting for 53.18% of the total capacity. There are 114 wind farms with a capacity of 13,766.08 MW, accounting for 24.64% of the total capacity. There are 169 photovoltaic power stations with a capacity of 11,219.566 MW, accounting for 20.08% of the total capacity. The distributed photovoltaic capacity is 751.567 MW, accounting for 1.34% of the total capacity. The total capacity of wind power and photovoltaic is 25,737.213 MW, accounting for 46.07% of the total installed

capacity of the integrated adjustment. There are two hydropower plants and 15 units with a capacity of 422.3 MW, accounting for 0.76% of the total capacity.

The analysis shows that on the power supply side, thermal power takes up the highest proportion of the total installed capacity of the power system, reaching about 50%, wind power takes up about 25%, photovoltaic about 20%, and other types of power take up about 5%. At the same time, the power grid structure of the region can be divided into three nodes: the northern region, the eastern region, and the southern region. In order to make the study more convenient, the power system of this model is set as three nodes, namely ABC, regardless of unit maintenance and other plans, and remove the power delivery unit. Finally, Node A has a total installed capacity of 4540 MW coal-fired unit, 1515 MW wind power unit and 1357 MW photovoltaic unit, accounting for 38.75% of renewable energy. Node B has a total installed capacity of 7186.4 MW of coal-fired units, 2651 MW of wind power, and 3409 MW of photovoltaic, with 41.16% of renewable energy. Node C has a total installed capacity of 4540 MW coal-fired units, 3409 MW wind power capacity, 3055 MW photovoltaic installed capacity, and 46.89% of renewable energy, as shown in Table 2. Table 3 shows the power transmission capacity between nodes A, B, and C. The power transmission capacity between nodes A and B is 3000 MW, the power transmission capacity between nodes B and C is 4000 MW, and the power transmission capacity between nodes A and C is 4000 MW. The line loss rate of each power transmission channel is 1%. Table 4 shows the parameters of different types of thermal power units. In this system, there are 36 units with 330 MW or less and 8 units with 600–660 MW. The climbing rate of both types of units is 2%. And different types of units have different minimum technical output, efficiency, start-stop cost, and kWh cost. The parameters rely on project data from the enterprise.

**Table 2.** Power generation unit parameters for each node in the basic scenario 0.

| Node | Coal-Fired Unit (MW) | Wind Power (MW) | Photovoltaic (MW) | Aggregate (MW) | Proportion of Renewable Energy (%) |
|---|---|---|---|---|---|
| A | 4540 | 1515 | 1357 | 7412 | 38.75 |
| B | 7186.4 | 2651 | 2376 | 12,213.4 | 41.16 |
| C | 4540 | 3409 | 3055 | 11,004 | 58.74 |
| Aggregate | 16,266.4 | 7575 | 6788 | 30,629.4 | 46.89 |

**Table 3.** Setting channel capability parameters for each node in the basic scenario 0.

| Channel | Channel Capacity (MW) | Loss (%) |
|---|---|---|
| Nodes A and B | 3000 | 1 |
| Nodes B and C | 4000 | 1 |
| Nodes A and C | 4000 | 1 |

**Table 4.** Technical and economic parameter Settings of each unit in basic scenario 0.

| Unit Capacity | 330 MW and Below | 600–660 MW |
|---|---|---|
| The Numbers | 36 | 8 |
| Rate of climb | 0.02 | 0.02 |
| Minimum technical output | 1/6 was 26%, 1/3 was 32%, 1/3 was 38%, and 1/6 was 48%; | 1/2 is 42%, 1/2 is 35%; |
| efficiency | 39% | 45% |
| Start-stop cost (ten thousand RMB) | 6 | 8 |
| kWh Cost (RMB/kWh) | 0.315 | 0.276 |

### 3.2. Analysis of Renewable Energy Output and Power Load

The power grid transmission lines in northwest China are mostly distributed in areas with a bad natural environment and locally strong winds sometimes occur. The maximum wind speed in the northern region is 28.8 m/s in March. The second is in April, July, and October at 27.1–27.7 m/s. Other months are 21.3~26.3 m/s. The maximum wind speed in the western region is 37.9 m/s in May, followed by 28.1 m/s in July, 26.0 m/s in August, and 25.6 m/s in June. Other months are 19.8~24.2 m/s. The maximum wind speed in the southern region is 25.2 m/s in May, followed by 24.0 m/s in April. In other months, the range is 16.7~21.9 m/s. The average annual solar radiation is 4950 MJ/m$^2$~6100 MJ/m$^2$, the annual sunshine hours are 2250~3100 h, and the sunshine percentage is 50~69%. On this basis, part of the output status of wind power and PV in this region is simulated as Figure 1.

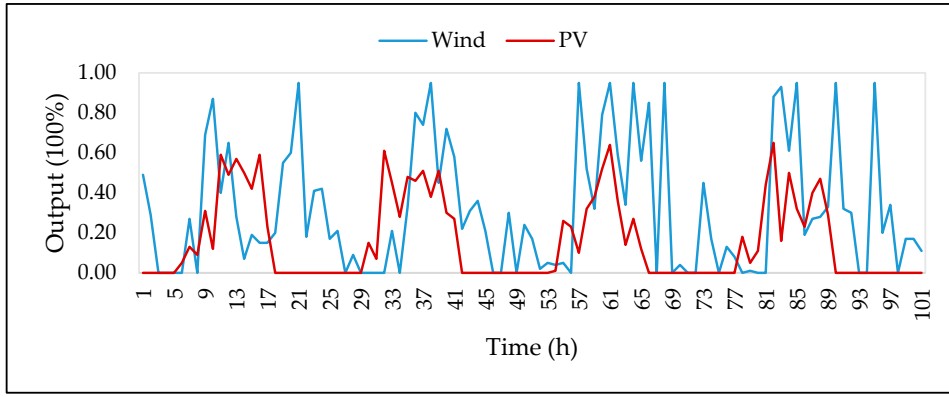

**Figure 1.** Wind power and PV output simulation.

In terms of electricity load, it is predicted that the industrial load in the region will maintain a high proportion in the future, and the proportion of electricity consumption in residential life and tertiary industries will increase. Due to the flexibility change of the simulation system in this paper, the power load will increase by 10% on the current basis. Using typical daily load curves, 8760 h of load data are simulated by uniform probability distribution. That is, the average value of each time point is set and the specified variance is given, so that the probability density function of random number distribution satisfies: $\mu$, $\sigma^2$.

$$f(x) = \frac{1}{2\sigma\sqrt{3}}, -\sigma\sqrt{3} \leq x - \mu \leq \sigma\sqrt{3},$$
$$f(x) = 0, else$$

Finally, the load curve of a certain day in summer and winter is generated as Figure 2:

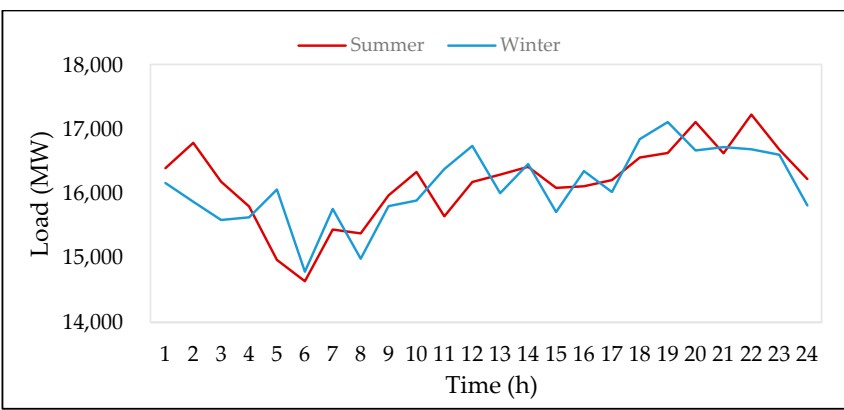

**Figure 2.** Electricity load simulation.

*3.3. Running Results of the Benchmark Model*

In this scenario, the result shown in Figure 3. Figure 3a shows the maximum amount of loss of load in this period, the result shows that the loss of load occurs at node A, Figure 3b shows the maximum amount of curtailment in this period, the result shows curtailment occurs at node B and C, Figure 3c shows the upward reserve provided by non-VRE units and Figure 3d shows the upward reserve provided by VRE units, those results show that both non-VRE and VRE provide upward reserve in this system. For the given period, the loss of load is 0.258% and the curtailment is 6.863%. It can be seen that the system does not appear serious load loss problem, but there is a renewable energy curtailment phenomenon.

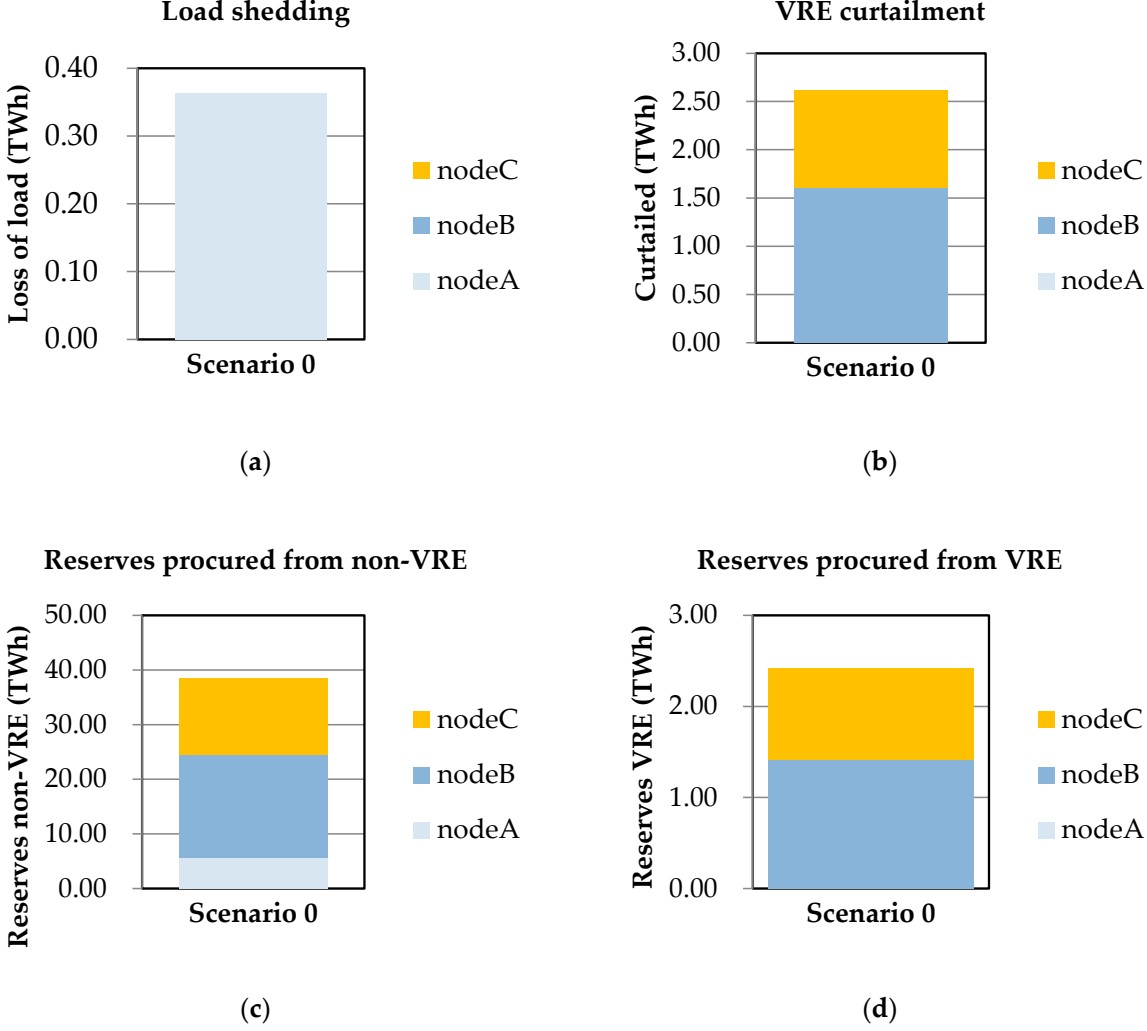

**Figure 3.** Running results of the benchmark mode. (**a**) shows the maximum amount of loss of load in scenario 0; (**b**) shows the maximum amount of curtailment in scenario 0; (**c**) shows the up-ward reserve provided by non-VRE units in scenario 0; (**d**) shows the upward reserve provided by VRE units in scenario 0.

## 4. Scenario Simulation of Multiple Types of Flexible Resources

*4.1. Analysis of Current Flexible Resource Endowment of the System*

The flexible resources of the power system are mainly obtained from four aspects: source, network, load, and storage. Therefore, the system described in this paper is judged from seven perspectives shown in Table 5.

**Table 5.** The flexible resources endowment of the system.

| Flexible Resource Endowment | High | Medium | Low |
|---|:---:|:---:|:---:|
| Capacity interconnection VS average demand | | ● | |
| Power generation climbing capability | | ● | |
| Electricity demand commensurate with renewable energy output | | | ● |
| Grid channel capacity | | ● | |
| Energy storage VS annual demand | | | ● |
| Geographic distribution VS renewable energy generation and demand | | ● | |
| Minimum demand VS renewable energy capacity | ● | | |

High indicates that the system performs well in this respect and can provide sufficient flexibility; the medium is normal; and low performance indicates poor performance, which is caused by the inflexibility of the system. Based on this, the scenario for improving the flexibility of this region was set.

According to the analysis results of scenario 0, the flexibility of the current system performance is not bad, and the loss of load and curtailment are not serious. As time goes on, more renewable energy sources are bound to be added to the future power system, namely increase renewable energy capacity. Due to the total installed capacity increased, so the corresponding load demand increased too. At the same time due to the increase in renewable energy and the increase in the load demand, certainly will require the system has more flexible resources to adjust, according to the analysis of Table 5, choose to reduce the thermal power unit minimum output, increase the energy storage, increase the channel capacity three ways, namely the scenarios 2–4, which shows in Table 6, to verify the flexible resources development status of the system.

**Table 6.** The scenario for improving the flexibility.

| Scenario | The Specific Content |
|:---:|:---:|
| Scenario 1 | Increase the share of renewable energy |
| Scenario 2 | Change thermal power minimum technical output |
| Scenario 3 | Increase the energy storage |
| Scenario 4 | Increasing channel capacity |

*4.2. Increase Renewable Energy Capacity*

Considering the successful performance of system flexibility, the installed capacity of renewable energy is increased as shown in the Table 7. The total capacity is increased by 20%, so the load is increased by 20%.

**Table 7.** The scenario for improving the sources of energy.

| Node | Coal-Fired Unit (MW) | Wind Power (MW) | Photovoltaic (MW) | Aggregate (MW) | Proportion of Renewable Energy (%) |
|:---:|:---:|:---:|:---:|:---:|:---:|
| A | 4540 | 3615 | 2957 | 11,112 | 59.14 |
| B | 7186.4 | 4251 | 3076 | 14,513.4 | 50.48 |
| C | 4540 | 5509 | 5155 | 15,204 | 70.14 |
| Aggregate | 16,266.4 | 13,375 | 11,188 | 40,829.4 | 60.16 |

The running results of this scenario are as Figure 4:

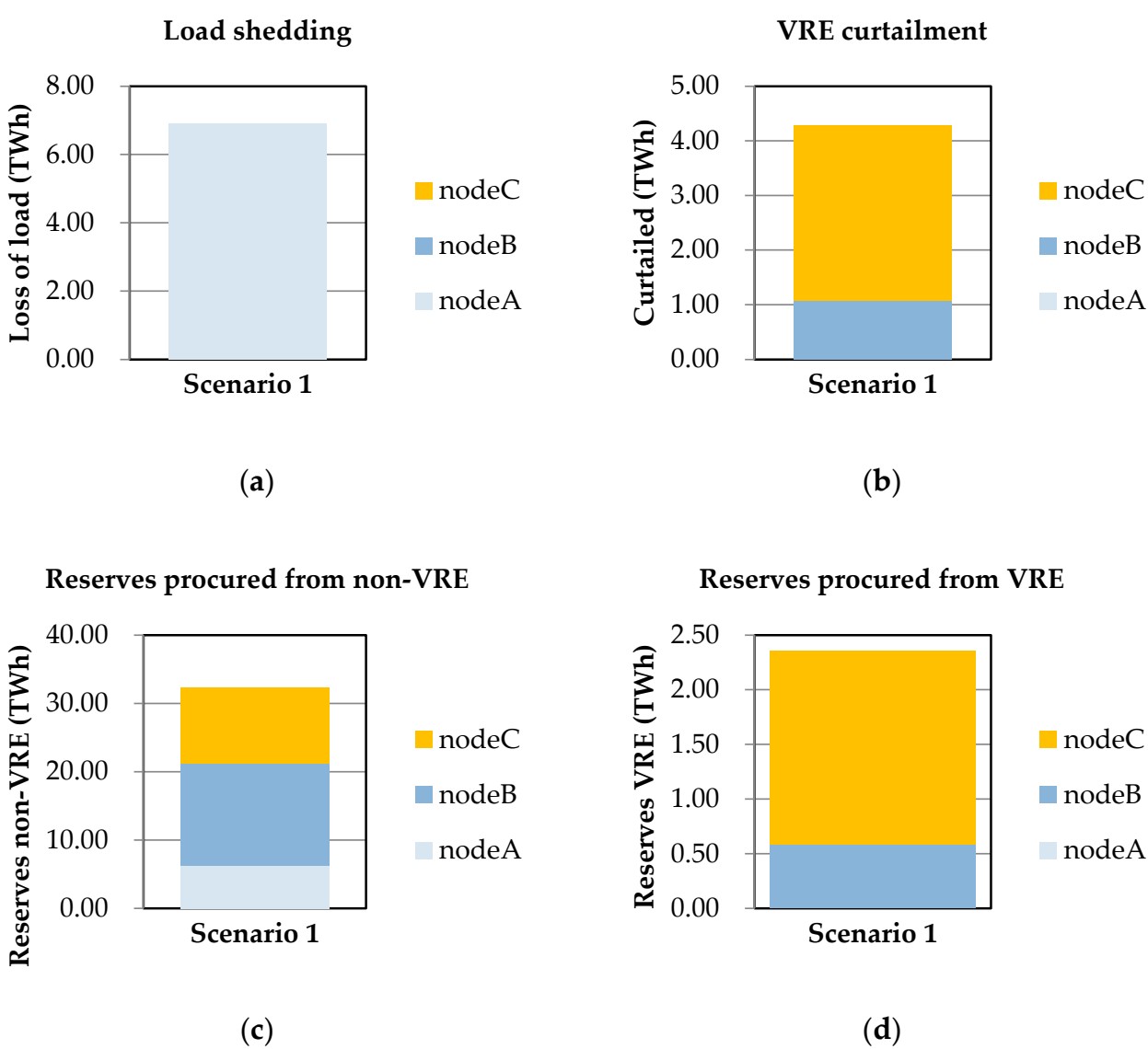

**Figure 4.** Running results of the scenario for increasing renewable energy capacity. (**a**) shows the maximum amount of loss of load in scenario 1; (**b**) shows the maximum amount of curtailment in scenario 1; (**c**) shows the up-ward reserve provided by non-VRE units in scenario 1; (**d**) shows the upward reserve provided by VRE units in scenario 1.

In this scenario, Figure 4a shows the maximum amount of loss of load in this period, which is higher than scenario 0; Figure 4b shows the maximum amount of curtailment in this period, which is also higher than scenario 0; Figure 4c shows the upward reserve provided by non-VRE units, which is lower than scenario 0; and Figure 4d shows the upward reserve provided by VRE units, which is almost same as scenario 0. For the given period, loss of load is 4.09%, curtailment is 8.042%, and the probability of insufficient reserve is 0. Flexibility performance decreased and needs more flexible resources to adjust.

### 4.3. Change the Minimum Technical Output of Thermal Power Units

Changing the minimum technical output of the thermal power units is conducive to the improvement of the peak adjustment capacity of the power system and the flexibility of the system. Therefore, in the case of increasing the proportion of renewable energy in scenario 1, the minimum technical output of the thermal power units is changed as shown in Table 8:

**Table 8.** The scenario for changing the minimum technical output of thermal power units.

| Unit Capacity | 330 MW and Below | 600–660 MW |
|---|---|---|
| Minimum technical output (before change) | 1/6 was 26%, 1/3 was 32%, 1/3 was 38%, and 1/6 was 48%; | 1/2 is 42%, 1/2 is 35%; |
| Minimum technical output (after change) | 1/6 was 24%, 1/3 was 30%, 1/3 was 36%, and 1/6 was 46%; | 1/2 is 40%, 1/2 is 33%; |

The running results of this scenario are as shown in Figure 5:

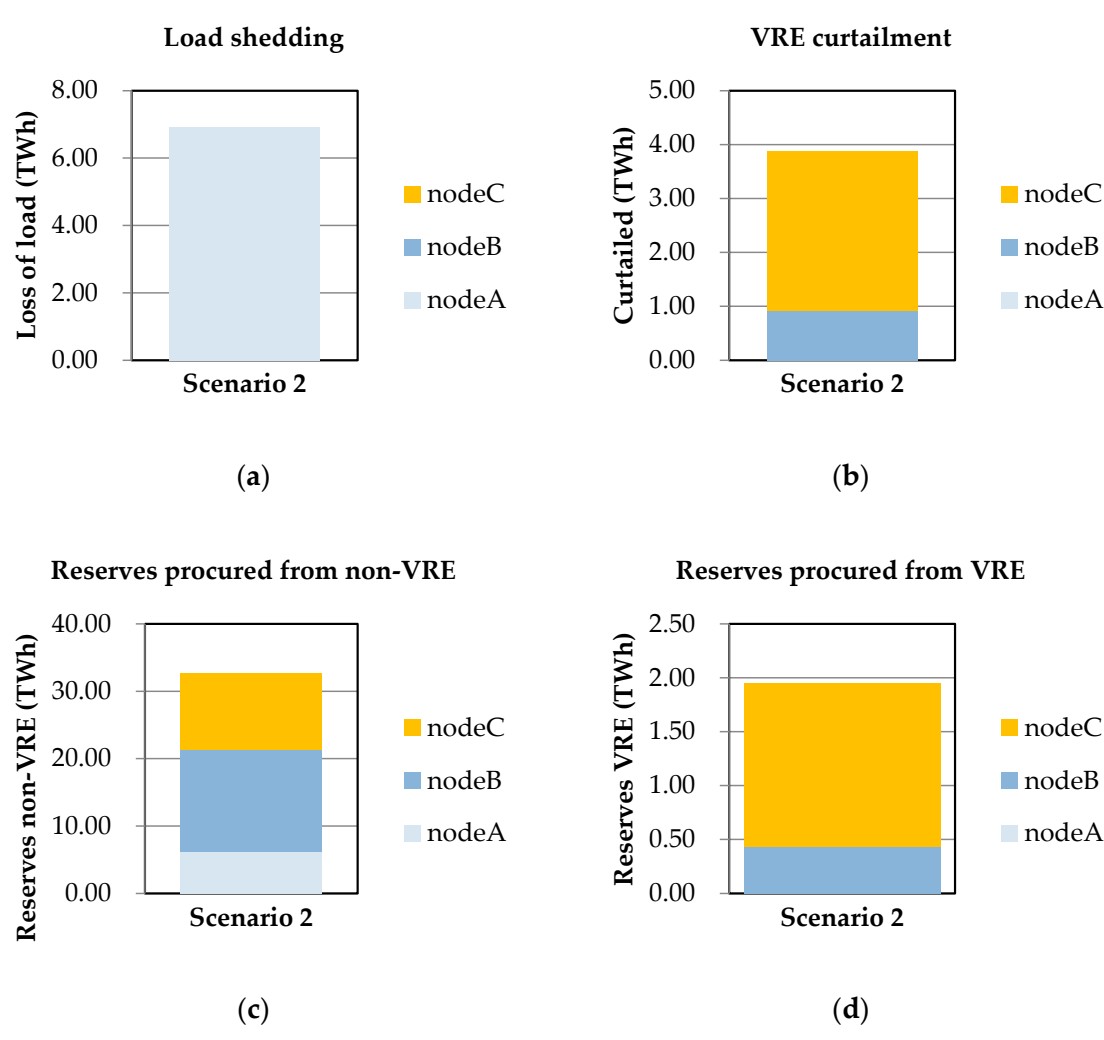

**Figure 5.** Running results of the scenario for changing the minimum technical output of thermal power units. (**a**) shows the maximum amount of loss of load in scenario 2; (**b**) shows the maximum amount of curtailment in scenario 2; (**c**) shows the up-ward reserve provided by non-VRE units in scenario 2; (**d**) shows the upward reserve provided by VRE units in scenario 2.

In this scenario, Figure 5a shows the maximum amount of loss of load in this period, which is almost the same as scenario 1; Figure 5b shows the maximum amount of curtailment in this period, which is lower than scenario 1; Figure 5c shows the upward reserve provided by non-VRE units, which is almost same as scenario 1; and Figure 5d shows the upward reserve provided by VRE units, which is lower than scenario 1. For the given period, the loss of load is 4.09%, curtailment is 7.227%, and the probability of insufficient reserve is 0. Compared with scenario 1, flexibility performance has been improved, but it still needs more flexible resources to adjust.

### 4.4. Increasing Energy Storage

Considering that the loss of load mainly occurs at node A, electrochemical energy storage of 500 MW is added to node A in this scenario, and the technical parameters of energy storage are as shown in Table 9:

**Table 9.** The scenario for increasing energy storage.

| Type | Charge and Discharge Efficiency (%) | Charge and Discharge Loss (%) | Rate of Climb (%) | Kwh Cost (Rmb/Kwh) |
|---|---|---|---|---|
| Electrochemical storage | 96% | 0.04% | 100 | 1.2 |

The running results of this scenario are as Figure 6:

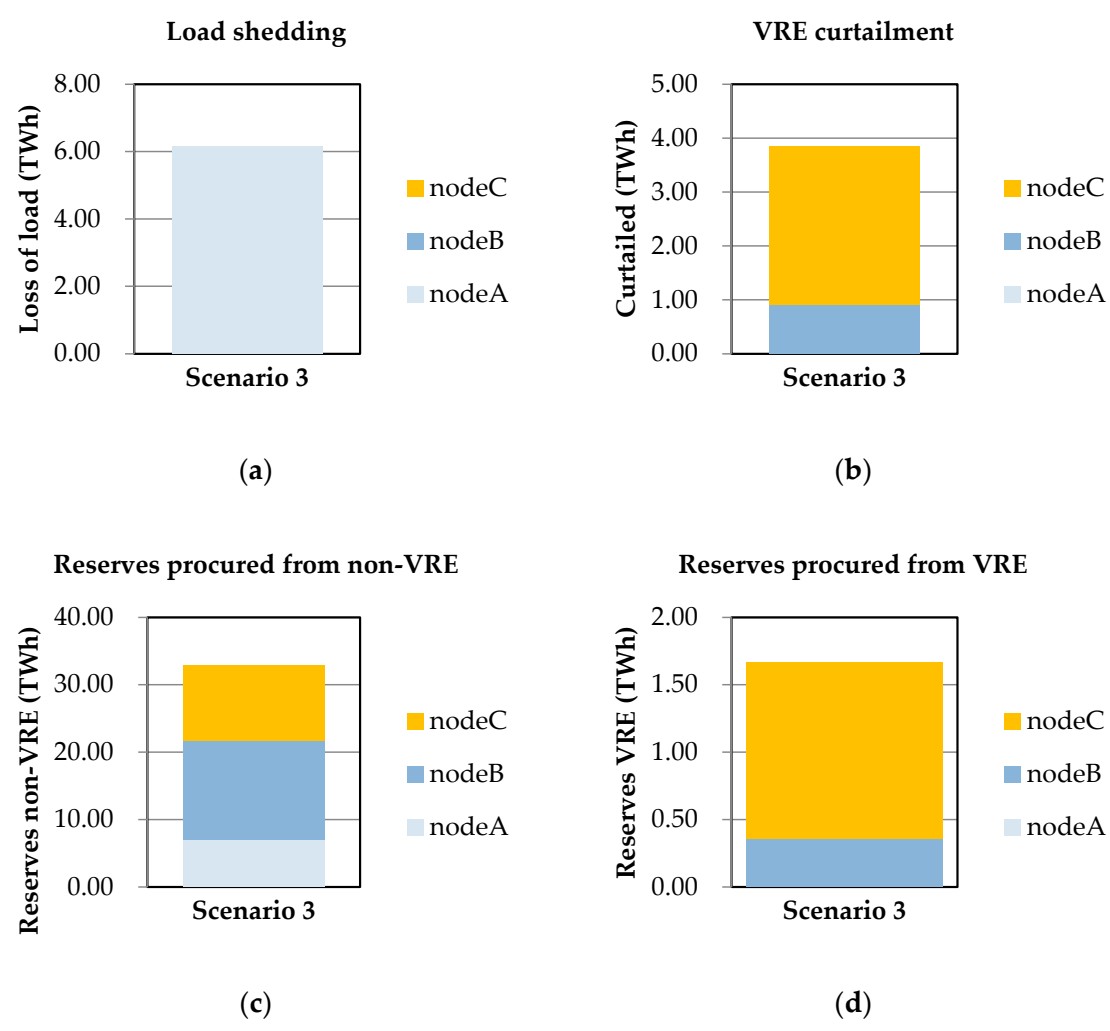

**Figure 6.** Running results of the scenario for increasing energy storage. (**a**) shows the maximum amount of loss of load in scenario 3; (**b**) shows the maximum amount of curtailment in scenario 3; (**c**) shows the up-ward reserve provided by non-VRE units in scenario 3; (**d**) shows the upward reserve provided by VRE units in scenario 3.

In this scenario, Figure 6a shows the maximum amount of loss of load in this period, which is lower than scenario 1; Figure 6b shows the maximum amount of curtailment in this period, which is lower than scenario 1; Figure 6c shows the upward reserve provided by non-VRE units, which is almost the same as scenario 1; and Figure 6d shows the upward reserve provided by VRE units, which is lower than scenario 1. For the given period, the

loss of load is 3.643%, curtailment is 7.189%, and the probability of insufficient reserve is 0. Flexibility performance has improved.

### 4.5. Increasing Channel Capacity

Considering that load loss mainly occurs at node A and wind abandoning and light abandoning mainly occurs at node C, the channel capacity between A–C nodes is increased by 300 MW in this scenario.

The running results of this scenario are as Figure 7:

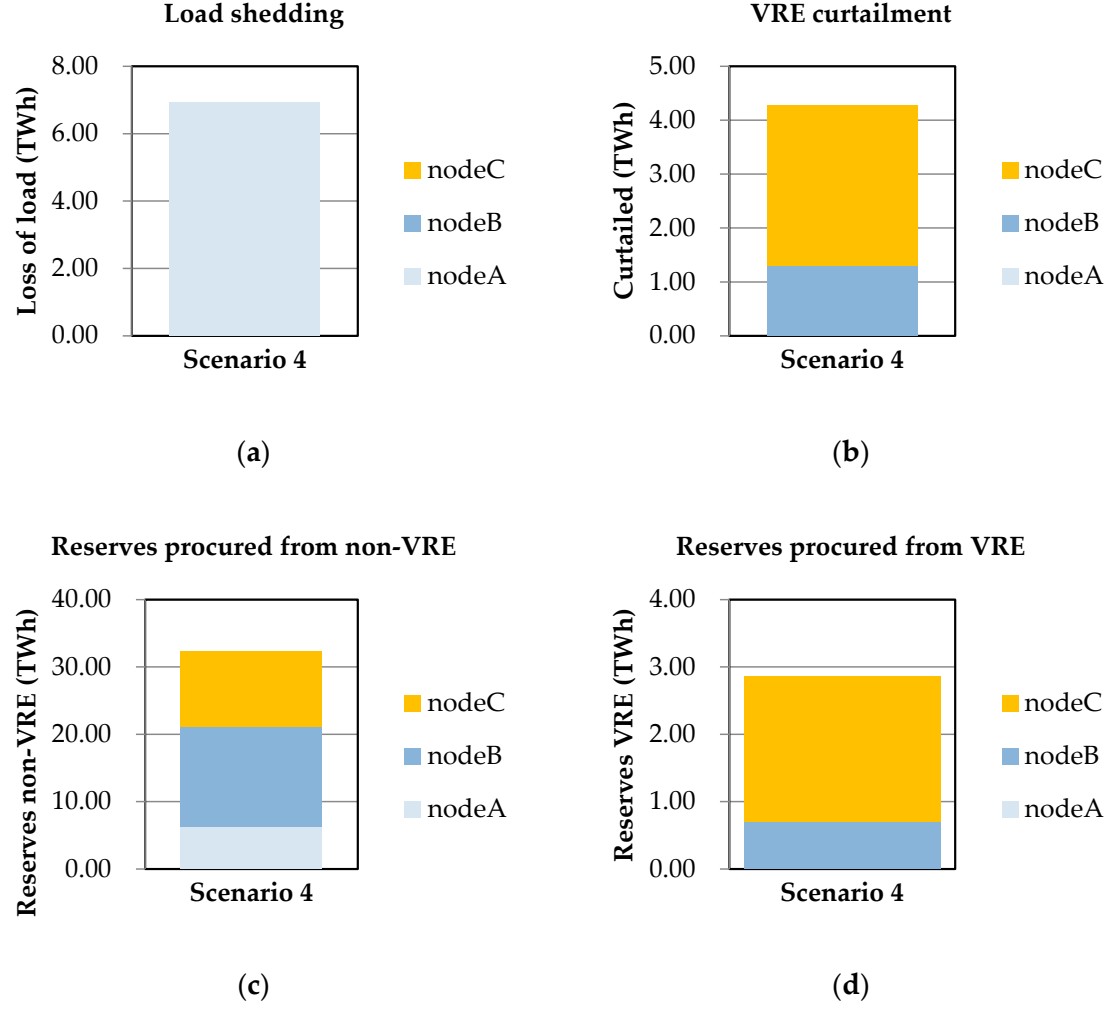

**Figure 7.** Running results of the scenario for increasing channel capacity. (**a**) shows the maximum amount of loss of load in scenario 4; (**b**) shows the maximum amount of curtailment in scenario 4; (**c**) shows the up-ward reserve provided by non-VRE units in scenario 4; (**d**) shows the upward reserve provided by VRE units in scenario 4.

In this scenario, Figure 7a shows the maximum amount of loss of load in this period, it is almost the same as scenario 1; Figure 7b shows the maximum amount of curtailment in this period, it is almost the same as scenario 1; Figure 7c shows the upward reserve provided by non-VRE units, it is almost the same as scenario 1; and Figure 7d shows the upward reserve provided by VRE units, it is higher than scenario 1. For the given period, the loss of load is 4.09%, curtailment is 8.042%, and the probability of insufficient reserve is 0. There was almost no improvement in flexibility.

Indicators evaluated by each scenario above are shown in Table 10. In addition, each scenario of the insufficient reserve is 0, so no more analysis will be done here.

**Table 10.** Indicators evaluated by each scenario.

| The Example Scenario | Loss of Load (%) | Curtailment (%) | The Total Cost (RMB) |
|---|---|---|---|
| Scenario 0 | 0.258 | 6.863 | 154,115 |
| Scenario 1 | 4.09 | 8.042 | 489,659 |
| Scenario 2 | 4.09 | 7.227 | 489,005 |
| Scenario 3 | 3.643 | 7.189 | 452,441 |
| Scenario 4 | 4.09 | 8.042 | 489,659 |

As Table 10 shows, in the system of more renewable energy and more electricity load, loss of load and curtailment are increased. In order to improve the flexibility of this system, this paper takes several flexible measures. As the results show, reducing the minimum technical output of thermal power units has an obvious effect on reducing curtailment but has no obvious effect on reducing loss of load. Increasing energy storage at node A has an obvious effect on reducing curtailment and loss of load. However, increasing the channel capacity of the A-C channel has no obvious effect on reducing curtailment and loss of load. So for this power system, if we want to reduce loss of load, can consider adding storage, if we want to reduce curtailment, can choose to add storage and to modify the thermal power unit to reduce the minimum technical output at the same time. Without considering the investment cost, increasing storage is the best of these measures.

*4.6. Sensitivity Analysis*

In order to verify the effectiveness of various flexibility improvement measures, sensitivity tests were conducted on various types of flexible resources, and the final results were as Table 11 shows:

**Table 11.** Sensitivity analysis of each scenario.

| Measures | Sensitivity Analysis | Loss of Load (%) | Curtailment (%) | Reserve Inadequacy | Total Cost (RMB) |
|---|---|---|---|---|---|
| Change the minimum technical output of thermal power units | No change | 4.09 | 8.042 | 0 | 489,659 |
| | A reduction of 2% | 4.09 | 7.227 | 0 | 489,005 |
| | A reduction of 4% | 4.09 | 6.466 | 0 | 488,392 |
| | A reduction of 6% | 4.09 | 5.745 | 0 | 487,804 |
| Increasing Energy Storage | No change | 4.09 | 8.042 | 0 | 489,659 |
| | Add energy storage 500 MW at node A | 3.643 | 7.189 | 0 | 452,441 |
| | Add energy storage 1000 MW at node A | 3.338 | 6.442 | 0 | 426,880 |
| | Add energy storage 1500 MW at node A | 3.086 | 5.802 | 0 | 405,759 |
| Increasing channel Capacity | No change | 4.09 | 8.042 | 0 | 489,659 |
| | Increase the channel capacity of nodes A and C by 300 MW | 4.09 | 8.042 | 0 | 489,659 |
| | Increase the channel capacity of nodes A and C by 600 MW | 4.09 | 8.042 | 0 | 489,659 |
| | Increase the channel capacity of nodes A and C by 900 MW | 4.09 | 8.042 | 0 | 489,659 |

The test result shows that compared with not taking any measures, reducing the minimum technical output efforts to reduce system curtailment has a positive effect, but

reducing the loss of load is not obvious, and the marginal contribution of it to reduce curtailment can keep a relatively stable level in a certain range. At the same time, this measure is beneficial to reduce the total cost of the system.

The test result shows that compared with not taking any measures, increasing energy storage has obvious positive effects on both reducing curtailment and reducing loss of load, but in a certain range, the marginal contribution of this measure to reduce loss of load gradually becomes smaller, and the marginal contribution of this measure to reduce curtailment also gradually becomes smaller. What is more, this strategy can reduce the total cost of the system more obviously.

The test result shows that compared with not taking any measures, increasing the channel capacity between nodes A and C has no obvious effect on improving the flexibility of the system. The results show that the insufficient flexibility of this system is not caused by the insufficient channel capacity.

## 5. Conclusions

The analysis shows that the current power system in a certain region in northwest China has good flexibility and can cope with the fluctuation of renewable energy and the change of load. However, if the proportion of renewable energy in the power system is increased in the future and the load continues to grow, the flexibility performance will decline, especially in node A, which is the northern region and will face the situation of loss of load. At the same time, there will be some curtailment of wind power output and PV output in all three nodes. According to the simulation results of this paper, it is suggested that adding storage at node A reduces the loss of load. Through the transformation of thermal power units, increase the energy storage, can reduce the curtailment of wind power and PV. Therefore, this paper puts forward the following suggestions for policymakers and power system planners in this region:

1.  Under the condition that the local electricity load demand is gradually increasing, more renewable energy can be considered to meet the load demand.
2.  In the context of more renewable energy, partial loss of load and more curtailment may occur in this region, especially at node A, where the loss of load is more serious, so it is necessary to transform the power system at the same time to improve the flexibility of the power system.
3.  The capacity of the power channels in the region is performing well; therefore, when upgrading the flexibility of the power system in the region, other measures should be prioritized over the construction of new power channels.
4.  If policymakers want to reduce the curtailment of wind power output and PV output in the system, they can consider the transformation of thermal power units to reduce the minimum technical output or add storage measures.
5.  If policymakers want to reduce both curtailment of wind power output and PV output and loss of load at node A, adding storage measures more suitable to be selected.

Increased storage is conducive to reducing the overall operation cost, but this paper has not considered the cost of the investment. Future work includes the following aspects. The first is to consider other types of flexibility measures, such as demand response, pumped storage, etc. The second is to consider the investment cost of flexibility transformation measures. And the third is to expand the study of the power grid to the study of cogeneration networks.

**Author Contributions:** Conceptualization, J.D.; Data curation, Z.C.; Investigation, Z.C.; Methodology, Z.C.; Resources, X.D.; Writing—original draft, X.D. All authors have read and agreed to the published version of the manuscript.

**Funding:** This research received no external funding.

**Institutional Review Board Statement:** Not applicable.

**Informed Consent Statement:** Not applicable.

**Data Availability Statement:** Not applicable.

**Conflicts of Interest:** The authors declare no conflict of interest.

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
