# Peer review of "The Influence of Multiple Types of Flexible Resources on the Flexibility of Power System in Northwest China"

_sustainability, doi:10.3390/su141811617_

Round 1

Reviewer 1 Report

1. Introduction:

The first paragraph of the literature review should be approved by mentioning some references. The style of other references should be updated according to he journal requirements.  

Clarify the research gap before talking about the contributions.

The basic scenarios (1,2 and 3) should be explained. 

The language quality of the manuscript need to be imporved.

There are some Chinese characters in the manuscript. They should be translated to English. 

2. Strategy

Why did you used this methods? what are its benefits and limitations?

3. Benchmark power system model construction

Mention the references of the data. 

4. Scenario simulation of multiple types of flexible resources 

Table 4-1, what does "In the" mean?

you can add a sensitivity analysis section to validate your results.

5- Conclusion

Add some policy implications or suggestions

Reviewer 2 Report

- try to use MW within the entire manuscript (instead of mw or other forms). The same for kWh (not KWH or other forms);

- fig. 1, fig. 2: measurement units should be provided;

- fig. 2: the legend should be written in English;

- fig. 3: some comments should be provided for each case (a - d). Also, all the acronyms should be explained at their first occurrence (VRE ??, SOP ??, ....);

- table 4-1: the title row is not clear. What is the meaning for "In the" (the 3rd column) ?

- table 4-5: measurement units should be provided (for the last column);

- at page 5 the authors are talking about 3 scenarios. In table 4-5 there are stipulated 4 scenarios ... It is recommended to describe all the analyzed scenarios in section 3.1;

- section 5: should be renamed "Conclusion" (singular);

- all the figures and tables should be referenced in the text and few comments (explanations) related to them should be provided. 

Reviewer 3 Report

The paper studies the key aspect affecting the flexible design the best mix of renewable energy sources in a given area.

All the following indicated aspects should be clarified and better explained in the manuscript.

Literature review

1.       The main contributions of the paper are clearly described. Nevertheless, from the current manuscript it is not grasp understanding the novelty of the work. The authors should better highlight the innovative aspects of their work in the manuscript.

System design

2.       The description of all the used variables could be grouped in a preliminary Table at the beginning of the problem statement.

3.       A formal definition of the addressed networks at the beginning of Section 2 is missing. Do the authors consider electric grids or even multi-carrier microgrids?

4.       Several recent scientific studies on energy storage modelling (e.g., https://doi.org/10.1109/TASE.2022.3148856, https://doi.org/10.3390/en12071231, documents that could be cited in the text), show that the use of energy storage systems improve the performance of modern grids. The Authors should comment this point, clarifying if they address both the electrical and the thermal devices.

Case study

5.       The control strategy used in the energy management strongly affects the results of the flexibility analysis. The authors could compare the results of their analysis with the scenario where an advanced and better-performing control strategy (such as model predictive control: e.g.,  https://doi.org/10.23919/ECC.2019.8796182, https://doi.org/ 10.17775/CSEEJPES.2020.02160, documents that could be cited in the text) is employed. The Authors should comment this point.

6.       There is no sensitivity analysis in the paper. Is it reasonable?

Conclusions

7.       Conclusions needs to be extended to present further implications for future research and many managerial insights based on the results of the study, as well as limitations.

Minor

8.       The authors should check that all the used acronyms are explained the first time they are used.

9.       Mainly the English is good and there are only a few typos.  However the paper should be carefully rechecked.

Round 2

Reviewer 3 Report

Previous comments and concerns have been sufficiently addressed. In the revised paper several improvements have been added.